# Emerging Role of microRNA Dysregulation in Diagnosis and Prognosis of Extrahepatic Cholangiocarcinoma

**DOI:** 10.3390/genes13081479

**Published:** 2022-08-19

**Authors:** Christian Prinz, Robin Frese, Mashiba Grams, Leonard Fehring

**Affiliations:** 1Medizinische Klinik 2, Helios Universitätsklinikum, 42283 Wuppertal, Germany; 2Lehrstuhl für Innere Medizin 1 der, University of Witten gGmbH, 42283 Wuppertal, Germany

**Keywords:** microRNA, cholangiocellular carcinoma, cholangioscopy, neovascularisation, angiogenetic pathways

## Abstract

Extrahepatic cholangiocarcinomas, also called bile duct carcinomas, represent a special entity in gastrointestinal tumors, and histological specimens of the tumors are often difficult to obtain. A special feature of these tumors is the strong neovascularization, which can often be seen in the endoluminal endoscopic procedure called cholangioscopy, performed alone or in combination with laserscanning techniques. The additional analysis of microRNA expression profiles associated with inflammation and neovascularization in bile duct tumors or just the bile duct fluid of these patients could be of enormous additional importance. In particular, the dysregulation of microRNA in these cholangiocarcinomas (CCA) was previously reported to affect epigenetics (reported for miR-148, miR-152), inflammation (determined for miR-200, miR-125, and miR-605), and chemoresistance (miR-200b, 204) in patients with cholangiocarcinoma. More importantly, in the context of malignant neovascularization, well-defined microRNAs including miR-141, miR-181, miR-191, and miR-200b have been found to be dysregulated in cholangiocarcinoma and have been associated with an increased proliferation and vascularization in CCA. Thus, a panel of these microRNA molecules together with the clinical aspects of these tumors might facilitate tumor diagnosis and early treatment. To our knowledge, this is the first review that outlines the unique potential of combining macroscopic findings from cholangioscopy with microRNA expression.

## 1. Cholangiocarcinoma—Clinical Background

Cancers of the biliary tract, also called cholangiocarcinomas (CCA, CCC, or CC) [1], are tumors of the extrahepatic (eCCA) or intrahepatic (iCCA) bile duct system. The worldwide mortality has increased dramatically during the past years, according to the WHO and other Health Organization databases, for different locations in the U.S.A., in Europe, but especially in East Asia [2]. CCA mortality has been reported to be higher in men than women worldwide, and Asian individuals were reported to have a high mortality (2.81 per 100,000 men in Japan). However, in the USA, an increased mortality was found between 2004 and 2014 for African American individuals (45%), followed by Asian (22%) and white (20%) individuals [3]. In Japan, the age-standardized incidence rates (ASRs) per 100,000 person-years for iCCA are higher than for eCCA (e.g., Japan: ASR for ICC, 0.95; ASR for ECC, 0.83) [4].

Based on the anatomical origin, CCAs can be classified as intrahepatic (iCCAs), perihilar (pCCAs), or distal (dCCAs) [1]. There are also some cases of combined hepatocellular cholangiocarcinoma [5]. The risk factors may vary depending on the location [6]. iCCAs are associated with overweight/obesity and chronic liver diseases involving cirrhosis and/or viral hepatitis; pCCAs are associated with primary sclerosing cholangitis; and dCCAs are associated with choledocholithiasis [7]. pCCAs and dCCAs often develop in the setting of prolonged inflammation and/or cholestasis, which contribute to carcinogenesis, especially in patients with primary sclerosing cholangitis (PSC). The unambiguous evaluation of the biopsies can be influenced by inflammatory lesions as well as the fibroblastic differentiation of the circumscribed tumors.

An integrative genomic analysis revealed four novel transcriptome-based molecular classes of extrahepatic cholangiocarcinoma and identified ~25% of tumors with actionable genomic alterations, which has potential prognostic and therapeutic implications [8]. One important factor is the fact that the chronic inflammation of bile ducts, especially present in male individuals suffering from primary sclerosing cholangitis and colitis, results in a very high risk of developing CCAs and may lead to early treatment of these patients, including liver transplantation.

## 2. Role of Cholangioscopy, Laser Scanning Microscopy, and Next Generation Sequencing (NGS) for Detection of Cholangiocarcinoma

Peroral cholangioscopy, i.e., endoscopy of the biliary tract introduced during endoscopic-retrograde cholangiography (ERCP), allows for a more precise diagnosis and treatment, and may thus be regarded as a true gold standard when it comes to the evaluation of unclear intrahepatic biliary filling defects and strictures [9]. Cholangioscopy can visualize bile duct strictures, and specific biopsy sampling is possible, so that an exact differentiation of bile duct adenomas, intraductal malignant tumors, polypoid lesions, biliary papilomatosis, or IgG4-dependent cholangiopathy is present [10]. Figure 1 shows a representative image of such bile duct cancer obtained via cholangioscopy.

An important previous study using reusable cholangioscopes by Prinz et. al investigated the use of a shorter mother-baby system (S-POCS) with regard to functionality and manageability in cases of suspicious bile duct strictures or fixed filling defects in the bile duct [10]. A representative aspect of such biliary cancers is seen in Figure 1A [10]. Currently, peroral videocholangioscopy (POCS) is performed with a single-use Mother-Baby-System (MBSS), in which an ultra-thin endoscope is inserted in a transpapillary manner through the instrumentation channel of a duodenoscope, due to the high frequency of cholangitis when cholangioscopy is performed. The current standard of care is the use of single-use cholangioscopes, provided by Boston-Scientific Inc. and termed as Spy-glass systems [11]. Additionally laser-scanning endomicroscopes are available Figure 1B.

## 3. MicroRNA in CCA: Emerging Role as Predictors of Cancer Presence

Numerous studies have proposed a wide spectrum of biomarkers at the tissue and molecular levels [12]. Recently, a new technique based on Next-Generation-Sequencing (NGS) of biliary tissue samples has been described in order to allow for a better prediction of the progression to malignancy by integrating next-generation sequencing (“BiliSeq”) with ERCP-obtained biliary specimens from pancreatic ductual adenocarcinoma (PDAC) or CCA candidates [13]. This study reported an improved detection of both biliary tract CCA and PDAC, with a sensitivity and specificity for malignant strictures of 73% in PDAC and 100% in CCA. In comparison, elevated serum values for the tumor marker CA19-9 and pathological evaluation had sensitivities of only 76% and 48% in PDAC and CCA, respectively. BiliSeq further improved the sensitivity of the pathological evaluation of the malignancy from 35% to 77% for biliary brushings and from 52% to 83% for biliary biopsies [13]. Interestingly, bile is the biological fluid with the highest diagnostic capacity for miRNA, followed by serum, tissue, and urine (AUC 0.95, 0.913, 0.846, and 0.745, respectively) [14].

The dysregulation of microRNA has been associated with cancer development and progression, underlining a potential as important clinical predictive markers. New evidence suggests a key role of these molecules in the development and progression of cancers of the biliary tract. MicroRNA dysregulation has been described in the fluid of the biliary tract obtained by ERCP and could be of clinical interest in patients with CCA. A recent meta-analysis of microRNA profiling showed 70 upregulated and 48 downregulated miRNAs in CCA [15]. Several studies have indicated that the altered expression of miRNAs could act as oncogenic or as a suppressor in the development and progression of CCA and could contribute as potential biomakers to clinical diagnosis and prognosis prediction [16]. The most prominent dysregulated pathways included phosphatidylinositol-3 kinases/Akt, mitogen-activated protein kinase, and Ras signaling pathways [15].

With their high prognostic values, miR-9 (AUC = 0.975) and miR-145 (AUC = 0.975) were shown to display a diagnostic capacity for CCA when compared with healthy individuals [17]. When comparing biliary specimens from patients with PSC-derived CCA and isolated PSC, miR-412, miR-640, miR-1537, and miR-3189 had a highly significant differentiating value [18]. miR-1537 in combination with CA19-9 resulted in higher diagnostic values than CA19-9 alone (AUC 0.91 versus 0.88; *p* > 0.05) [14].

In CCA patients, the serum levels of miR-21 ([19,20,21]), a well-known oncogenic miR in numerous other tumors as well such as breast cancer or gastric cancer, were found to be increased compared with healthy individuals, positively correlating with the clinical stage and poor survival. However the translation of this miRNA into clinical practice should be treated carefully, since it is usually increased in the serum and/or plasma of patients with many other cancers [22,23,24]. Other more specific microRNAs involved in a special pathogenesis might thus help to discriminate malignant tumors from other alterations and may be used in clinical practice when histological evaluation fails.

## 4. MicroRNA Dysregulation and Chemosensitivity of CCA

MicroRNA expression may also help clinicians determine the chemosensitivity of biliary tract cancer when endoscopic therapy with radiofrequency ablation is performed in malignant lesions of the biliary tract or in pancreatic cancer. Recent studies highlight the dramatic benefit of performing a concomitant chemotherapy during endoscopic radiofrequency ablation of biliary tract cancer. In that study, endoscopic radiofrequency ablation plus a novel oral 5-fluorouracil compound versus radiofrequency ablation alone for unresectable extrahepatic cholangiocarcinoma proved to be highly effective and almost doubled the life expectancy in these patients [25]. Future studies should thus target a quick assay to determine the mircroRNA expression profile associated with the chemosensitivity of such tumors before local ablation can be performed [26,27,28,29,30]. When NGS sequencing was performed in biopsies obtained from patients with biliary tract cancer, relevant genomic alterations were identified in 20 (8%) patients. Two patients with *ERBB2*-amplified cholangiocarcinoma received a trastuzumab-based regimen and had a measurable clinic-radiographic response [13].

In regard to the signal transduction pathways involved in CCA, the dysregulation of specific microRNA was reported to affect the cell cycle [31,32], epigenetics (miR-148, miR-152) [33,34,35], inflammation (determined for miR-200, miR-125, and miR-605) [36], as well as chemoresistance (miR-200b, 204) [37,38]. Specifically, for epigenetic regulation, miR-148a, miR-148b, and miR-152 seem to be involved in a feedback loop with DNA methyltransferase (DNMT) 1 in some tumor types [39,40,41]. While the miR-148/-152 family post-transcriptionally regulates DNMT1 expression, the miRNA promoter methylation is reduced. Additionally, epithelial to mesenchymal transition (*EMT*), migration, and invasion was reported for microRNAs [42], allowing one to explain the strong fibroblastic transition of these tumors, which makes it difficult to evaluate the biopsies during a histopathological evaluation.

## 5. Role of microRNA Dysregulation in Neovascularization

Neovascularization refers to the process whereby new blood vessels are formed from existing ones following endothelial cell proliferation and migration [43]. Due to the high cell turnover, many tumors have a high demand of oxygen, and tumor growth is accelerated by neovascularization [44]. Therefore, angiogenesis can be a prognostic factor for tumor growth and a potential therapeutic target [45].

MicroRNAs that can be regarded as being of special importance in the context of chronic inflammation and neovascularization include miR141, miR-181, miR-191, and miR-200b, which have also been found to be dysregulated in cholangiocarcinoma [46,47]. Previous works have emphasized that miRNAs play key roles in the regulation of cellular processes, such as increased proliferation and vascularization [46,47]. A key molecule in this context is the continuous, sustained levels of VEGF-A.

Recently, it was also reported that not only are pro-angiogenic factors important, but the downregulation of anti-angiogenic factors is as well. A research group recently reported that human adenocarcinomas commonly harbor mutations in the *KRAS* and *MYC* proto-oncogenes and in the *TP53* tumor suppressor gene, and that these genetic lesions can be regarded as being potentially pro-angiogenic, as they increase the production of vascular endothelial growth factor (*VEGF*). In this context, enhanced neovascularization has previously been identified as being correlated with the downregulation of anti-angiogenic thrombospondin-1 (*Tsp1*) and related proteins, such as connective tissue growth factor (*CTGF*), and both *Tsp1* and *CTGF* are predicted targets for repression by the miR-17-92 microRNA cluster, which was upregulated in colonocytes coexpressing K-Ras and c-Myc. In fact, microRNA dysregulation and even miR-17-92 knockdown with antisense 2′-O-methyl oligoribonucleotides partly restored *Tsp1* and *CTGF* expression, while miR-17-92–transduced cells formed larger, better perfused tumors [33].

Furthermore, it was shown that in the context of CCA, miR-320 downregulation correlated negatively with Neuropilin-1 (*NRP-1*). In animal models, the depletion of NRP-1 inhibited the activation of *VEGF/VEGFR2, EGF/EGFR,* and *HGF/c*-Met pathways stimulated by respective ligands [34]. The downregulation of these growth factors suppressed tumorigenesis, tumor growth, and lung metastasis by inhibiting cell proliferation and tumor angiogenesis. miR-320 negatively regulated the expression of *NRP-1* by binding to the 3’-UTR of the *NRP-1* promoter, thereby promoting tumor angiogenesis [34]. On the other hand, miR-101 inhibited cholangiocarcinoma angiogenesis by the direct targeting of the *VEGF* mRNA 3’ untranslated region and by the repression of *VEGF* gene transcription through the inhibition of COX-2 [35].

In a mechanistic study by Leng et al., the group showed that miR-490-3p attenuated cell migration and angiogenesis in CCA cells by silencing *Akirin2*, thereby inducing angiogenesis by increasing the expression of *VEGFA* by activating the IL-6/STAT3 signaling pathway [36]. In addition, the Angiotensin II type 1 (AT1) receptor blocker (ARB) Telmisartan has been discussed as a potential drug to inhibit cancer cell proliferation in CCA. Several miRNAs were significantly altered following telmisartan treatment in vitro, e.g., miR-3178, which acted as a tumor suppressor to inhibit proliferation, migration, invasion, and angiogenesis and which promoted the apoptosis and G1 phase, or miR-425-5p, which has recently been reported as being upregulated and promoting tumorigenesis in various cancer types [37].

## 6. Conclusions

From a clinical point of view, extrahepatic cholangiocarcinoma often presents with suspicious bile duct strictures and cholestasis, but histopathological approval with consecutive surgical treatment is often delayed due to difficult tissue sampling and the poor outcome of the histo-pathological evaluation of the fibrotic tumor samples. Thus, the combination of innovative cholangioscopy, laser scanning microscopy, and new, easily obtainable biomarkers might help facilitate tumor diagnosis at early stages. In this context, the microRNA expression profiling of bile fluid or tumor specimens in patients with unclear biliary tumors or biliary tract stenosis might improve the detection of cancer. Based on numerous studies, a panel of microRNAs listed in Table 1 has been evaluated and seems to be of special value here. Interestingly, all of the molecules are associated with increased neovascularization. Specific MicroRNAs that should be regarded as being of special importance in this context are, e.g., miR141, miR-181, miR-191, and miR-200b, which have also been found to be highly dysregulated in cholangiocarcinoma [46,47]. Studies have further emphasized that miRNAs play key roles in increased cellular proliferation [46,47]. In accordance with the effects, microRNA dysregulation and even the knockdown of the miR-17-92 cluster with antisense 2′-O-methyl oligoribonucleotides partly restored the expression of the anti-angiogenic factors *Tsp1* and *CTGF*, while miR-17-92–transduced cells formed larger, better perfused tumors [33]. Thus, microRNA deregulation might not only induce neovascularization, but might also suppress anti-angiogenic factors. In conclusion, microRNA deregulation may be a key factor in the molecular process of CCA. The evaluation of the dysregulation of defined microRNAs might facilitate an earlier and more successful diagnosis and treatment of extrahepatic CCAs.

## Figures and Tables

**Figure 1 genes-13-01479-f001:**
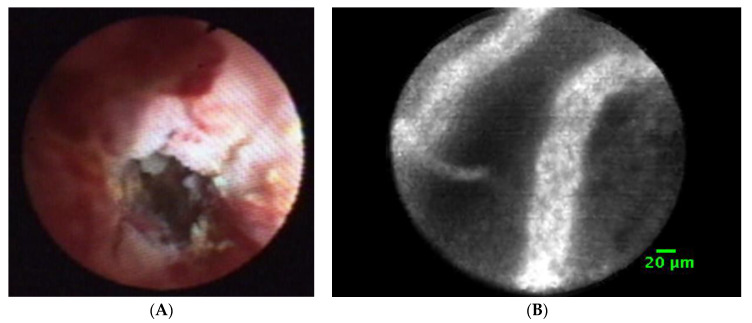
(**A**) Neovascularization detected by cholangioscopy. A peroral endoscope with small lumen (3.2 mm) is introduced into the bile duct, and an image of the intraluminal structure of the bile duct tumor is presented. The tumor shows a narrow stenosis, increased neovascularization, but also a fibrotic tumor. The width of the inner lumen ranges around 2 mm, and it cannot be passed by the endoscope. Details of the endoscopic approach are given in reference [10]. (**B**) Endomicroscopic laser-scanning microscopic picture of extrahepatic cholangiocarcinoma using laser scanning tips and visualization of fluorescent signaling following the injection of fluorescein in CCA patients. A strong vessel formation (approx. 20 µm in size) can be detected in the tumor, typical for CCA.

**Table 1 genes-13-01479-t001:** Overview of dysregulated microRNAs involved in CCA.

MiRNA Dysregulation in Biliary Tract Cancers and Potential Targets	
miR-9	useful diagnostic marker for CCA [17], miR-9 induces cell arrest and apoptosis of carcinoma cells via CDK 4/6 pathway [48]
miR-141	highly overexpressed in CCA cells, correlating with multifocal cholangiocarcinoma and vascular invasion [49]
miR-145	useful diagnostic markers for biliary tract cancer [17]
miR-181	therapeutic targets for the treatment of various diseases. Targets *MKP-5* and regulates *p38 MAPK* activation [50], and regulates *NDRG2* to influence carcinogenesis and metastasis [51]
miR-191	miR-191 acts as a potential therapeutic target [52]
miR-200b	highly overexpressed in CCA cells, inhibition of miR-200b is associated with sensitivity to gemcitabine [49]
miR-412	differed significantly between patients with PSC and PSC/CCA [18]. miR-412-5p targets *Xpo1* to regulate angiogenesis [53]
miR-640	differed significantly between patients with PSC and PSC/CCA [18]. miR-640 acts via NF-kappaB and WNT signaling pathway [54]
miR-1281	differed significantly between patients with PSC and PSC/CCA, also combining miR-1537 with CA19-9 resulted in higher diagnostic values than CA19-9 alone [18]. miR-1281 is a *p53*-responsive microRNA that impairs the survival of sarcoma cells upon ER stress [55]
miR-3189	differed significantly between patients with PSC and PSC/CCA [18]. miR-3189-3p mimics the effects of S100A4 siRNA upon the inhibition of proliferation and migration of gastric cancer cells by targeting *CFL2* [56]
miR-204	negatively regulates *Mcl-1* or *Bcl-2* expression and facilitates chemotherapeutic drug-triggered apoptosis [41]
miR-1537	Downregulated in CCA [13]
miR-152	Inhibits migration, invasion, and *EMT* in Intrahepatic Cholangiocarcinoma [38]
miR-148a	IL-6 can regulate the expression of methylation-dependent tumor suppressor genes by modulation of miR-148a [39]
miR-17-92	expression of the miR-17-92 cluster is regulated by IL-6/Stat3, a key oncogenic signaling pathway pivotal in cholangiocarcinogenesis [33,40]
miR-320	Promotes tumor angiogeneseis in CCA by negatively regulating expression of Neuropilin-1 [34]
miR-101	inhibits cholangiocarcinoma angiogenesis by direct targeting of VEGF [35]
miR-490-3p	attenuates cell migration and angiogenesis in CCA by silencing Akirin2 [36]
miR-122	lncRNA-UCA1 promoted metastasis of bile duct carcinoma cells by regulating miR-122/*CLIC1* and activating the ERK/MAPK signaling pathway [57]
miR-122-5p	miR-122-5p inhibits proliferation and invasion of bile duct carcinoma cells and promotes cell apoptosis by targeting ALDOA expression [58]
miR-31	miR-31 might be a biomarker that reflects IL-6 expression in bile duct cancer tissues and predicts poor prognosis. Expression in bile duct cancer cells is significantly higher than that in normal bile duct epithelial cells and is significantly associated with shorter survival [59]
miR-16	miR-16 is downregulated in patients with CCA and can be used as biomarker to differentiate from pancreatic ductal adenocarcinoma and benign disease [60]
miR-877	miR-877 is upregulated in CCA and could be used as a diagnostic classifier for distal bile duct tumors [60]
miR-329	miR-329 inhibits bile duct cancer progression through translational repression of laminin subunit β 3, which leads to a suppression of epithelial-to-mesenchymal transition and lymph node metastasis [61]

## Data Availability

Not applicable in this review.

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
