# Peer review of "Emerging Role of microRNA Dysregulation in Diagnosis and Prognosis of Extrahepatic Cholangiocarcinoma"

_genes, 2022, doi:10.3390/genes13081479_

Round 1

Reviewer 1 Report

This review paper aims to reference miRNAs as tools for diagnosing and prognosis of CCA. Although I am not an expert in CCA, in my opinion, the authors could not convince me of this importance. Moreover, being a review in miRNAs it should be given more credit to this subject, and it is only present in 2 sections (3 and 5). Other issues, are (1) CCA abbreviation - Cholangiocarcinoma abbreviation is often CCA and no CC or CCC: doi: 10.1038/s41575-020-0310-z ; doi: 10.1136/gutjnl-2022-327099; doi:  10.1016/j.jhep.2021.01.035; (2) the conclusions, there are no conclusions not even future perspectives. Therefore, in my opinion, the authors should revise their manuscript and focus more on miRNA and also add some missing miRNAs that were already described for CCA.

Author Response

Many thanks for the detailed review. Please find attached our response in a separate table. 

Greetings, Christian Prinz and Leonard Fehring

Reviewer 2 Report

In this review by Prinz and colleagues, the authors describe potential implications of miRNome dysregulation for diagnostic workup and management of eCCA. eCCA comprises understudied tumour entities (particularly relative to intrahepatic CCA, iCCA) so a review on this topic is highly welcome. Rather than generically listing individual miRNAs that are dysregulated, the authors contextualize their description of miRNAs in ‘real world’ clinical challenges, including earlier diagnosis, poor tissue recovery by endoscopic procedures, and treatment decision-making. This is a solid review that should be well received by the clinical communities in this area.

Some comments for improvement:

-       Although the article deals with eCCA, the introduction and clinical background provides information on CCA in general (incorporating statistics for iCCA and eCCA combined together). Please provide information specific to eCCA or, alternatively, point out differences between iCCA and eCCA after providing a general background to CCA. Key issues to address include differences in incidence and mortality rates between iCCA and eCCA, including reasons as to why these differences exist. Relevant research articles in support of these points include, but are not limited to, the following: Izquierdo-Sanchez et al. Journal of Hepatology 2022, Bertucio et al. Journal of Hepatology 2019 (already referenced in this manuscript), Florio et al. Cancer 2020, Clements et al. Journal of Hepatology 2020.

-       “According to transcriptomic profiles, the ‘inflammation’ (38%) and ‘proliferation’ (62%) of subtypes were previously identified and reported to be differentially enriched with activation of the pro-inflammatory as well as oncogenic pathways [3].” Reference 3 does not correspond to transcriptomic subgroups of eCCA. Based on the reported percentages, I suspect the authors may have intended to refer to Sia et al. Gastroenterology 2013? If so, this is an inappropriate reference as this study focused on iCCA. This same group published a transcriptomic subclassification for eCCA in 2020 that is a valid reference in this context (Montal et al. Journal of Hepatology 2020). Please edit the text accordingly with the appropriate references.

-       Section 3, paragraph 2, sentence 2: “a certain diagnostic capacity for CCC when compared with healthy individuals” is quite vague, please report a classification performance metric such as AUC/precision/sensitivity/specificity or similar to improve clarity for the reader.

-       Paragraph 4 in section 3 briefly mentions functional biological roles of miRNA and does not fit the content of the section (“MicroRNA in CCC: emerging role as predictors of cancer presence”), so it likely should be moved the following section.

-       In the context of this review, clear reasons are given for why miRNA impact on diagnosis and chemoresistance are important during diagnostic workup. However, no reason is given for the importance of miRNAs on neovascularization in the context of diagnostic workup described in this review. Please provide a clinical rationale for why this is important at the beginning of section 5.

Typos, grammar, etc.

Abstract: “diagnose” should be “diagnosis”.

Section 2, sentence 1: “diagnose” should be “diagnosis”.

Section 2, last sentence: “IgG4 dependent” should be “IgG4-dependent”.

Section 2, paragraph 3, first sentence: “cholangioscope” should be “cholangioscopes”.

Please italicize Klebsiella.

Section 3, paragraph 2, first sentence: “Micro RNA” should be “MicroRNA”.

Section 3, paragraph 2, first sentence: “during ERCP could” should be “during ERCP and could”.

Section 3, paragraph 3, first sentence: “was found” should be “were found”.

Section 3, paragraph 3, first sentence: “[4] and [5]” should be “[4-5]”.

Section 3, paragraph 3, second sentence: “[6, 7] and [8]” should be “[6-8]”.

Section 5 heading: “ROLE” should be “Role”.

Section 5, paragraph 1, first sentence: “(as being non-coding (nc-)RNAs)” is unnecessary, please delete.

Section 5, paragraph 2, first sentence: “anti-angiogenetic” should be “anti-angiogenic”.

Please italicize gene names (e.g. KRAS, TP53) throughout the manuscript.

Section 6, paragraph 1, first sentence: “might facilitate to predict cancer presence” should be “might improve detection of cancer” or similar.

Section 6, paragraph 1, second sentence: “:” should be “.” at the end of the sentence.

Section 6, paragraph 1, third sentence: “(as being non-coding (nc-)RNAs)” is unnecessary, please delete.

Section 6, paragraph 1, second-last sentence: “anti-angiogenetic” should be “anti-angiogenic”.

Author Response

Many thanks for the very detailed review that really helped us to improve the overall quality. Please find our comments and changes in a separate table. greetings, Christian Prinz

Reviewer 3 Report

    The reviewer feels that the manuscript does not justify the title "Emerging Role of microRNA Dysregulation in Diagnosis and Prognosis of Extrahepatic Cholangiocarcinoma”. The reviewer advises the authors to focus and provide a comprehensive review on the importance of miRNAs in the diagnosis and prognosis of Cholangiocarcinoma. Apart from this, there are several concerns/comments, please find them below: 

1.   The abstract should be re-written, the reviewer thinks that a lot of focus has been given on the challenges of collecting histological samples, which might be true but is not the focus of the review.

2.    The reviewer fails to understand the relevance of the Section 2, “Role of cholangioscopy and laser scanning microscopy for detection of cholangio-carcinoma”. Given the title of the review this sub-section is misplaced. In the introduction the authors can briefly say about the collection of histological samples and their inherent challenges.  

3.  The same is true for Section 3 “MicroRNA in CCC: emerging role as predictors of cancer presence” first paragraph “Recently, a new technique has been…from 52% to 83% for biliary biopsies”, how is it relevant to the review. The section 2 and 3 can be clubbed together and briefly included in the introduction/background part.

4.   The authors have written statements without any details or references; for example, “Interestingly and potentially of greater specificity in the context of CCC, dysregulation of other microRNA was also reported to affect the cell cycle, epigenetics (miR-148, miR-152), inflammation (determined for miR-200, -125, and miR-605), as well as chemo-resistance (miR-200b, 204). Also, epithelial to mesenchymal transition (EMT), migration and invasion was reported for these microRNAs, allowing to explain the strong fibro-blastic transition of these tumors, which makes it difficult when evaluation the biopsies during histopathological evaluation”. The above phrase has no detailed information about the study, their limitation etc. MiRNAs affecting epigenetics is a vague statement.

5.    Section 4 – “MicroRNA dysregulation and chemosensitivity of CCC” has no information about any miRNAs involved in chemosensitivity and therapy resistance. The entire manuscript fails to convey the idea that why miRNAs are important in the diagnosis and prognosis of Cholangiocarcinoma. The authors can also include how miRNAs can be useful in diagnosing different types of cholangiocarcinoma.

6. Several key references are missing, PMID: 31106658, 29860474, 31443224, 30887508, 30121648 and many more.

7. The reviewer appreciates section 5, all the other sections relating to importance of miRNAs in different aspects of Cholangiocarcinoma should be as comprehensive if not more.

8.   The authors should perform a comprehensive review of studies reporting miRNA dysregulation in cholangiocarcinoma, miRNAs as biomarkers, in therapy resistance, in-vivo studies if any and potential of miRNAs as a therapeutic approach in cholangiocarcinoma and the inherent challenges associated with it. 

9.  The study should be different and an advancement over recent articles and reviews; for example, “Dysregulation of microRNA in cholangiocarcinoma identified through a meta-analysis of microRNA profiling” by Likhitrattanapisal S. et al., World J Gastroenterol. 2020 Aug 7; 26(29): 4356–4371 and “The Role of microRNAs in Cholangiocarcinoma” by Shi T. et al., Int J Mol Sci. 2021 Jul; 22(14): 7627, respectively.

10. The conclusion should be improved with author’s point of view and future perspectives.

Author Response

Many thanks for the really detailed comments that helped us to restructure the entire manuscript. We hope that with the changes made, the ms is now suitable for publication. Greetings, Christian Prinz

Round 2

Reviewer 1 Report

Nothing to point out

Author Response

Christian Prinz

Helios Universitätsklinikum Wuppertal

Conc: Reply to the questions raised by Reviewer 3 in the R2 process

MS: 1795468

At first, we wish to thank you for your positive response regarding our manuscript 1795468 entitled „Emerging Role of microRNA Dysregulation in Diagnosis and Prognosis of Extrahepatic Cholangiocarcinoma.“ Attached you find our updated version of the manuscript with all the minor revisions addressed.

Reviewer 2 Report

The authors have addressed all my comments and I fully support it for publication.

Author Response

Christian Prinz

Helios Universitätsklinikum Wuppertal

Conc: Reply to the questions raised by Reviewer 2 in the R2 process

MS: 1795468

At first, we wish to thank you for your positive response regarding our manuscript 1795468 entitled „Emerging Role of microRNA Dysregulation in Diagnosis and Prognosis of Extrahepatic Cholangiocarcinoma.“ Attached you find our updated version of the manuscript with all the minor revisions addressed.

Reviewer 3 Report

1. My second last comment was for the authors to detail how their manuscript is different and an improvement over the recent review, for example, The Role of microRNAs in Cholangiocarcinoma” by Shi T. et al., Int J Mol Sci. 2021 Jul; 22(14): 7627. This concern has not been answered by the authors.

2. Minor grammatical errors/spell check required.

Author Response

(The authors gave the same response as above.)
